 RESEARCH ADVANCE

# Independent validation of transgenerational inheritance of learned pathogen avoidance in *Caenorhabditis elegans*

**Aalimah Akinosho†, Joseph Alexander†, Kyle Floyd, Andres Gabriel Vidal-Gadea***

School of Biological Sciences, Illinois State University, Normal, United States

## eLife Assessment

This **valuable** study concerns a model for transgenerational epigenetic inheritance, the learned avoidance by *C. elegans* of the PA14 pathogenic strain of *Pseudomonas aeruginosa*. A recent study questioned whether transgenerational inheritance in this paradigm lacks robustness. The authors of this study have worked independently of the group that reported the original phenomenon and also independently of the group that challenged the original report. With **solid** data, this study independently validates findings previously reported by the Murphy group, confirming that the paradigm is reproducible elsewhere. The reviewers also appreciated the information on reagent sources used by different groups. The present study is therefore of broad interest to anyone studying genetics, epigenetics, or learned behavior.

*For correspondence:
avidal@ilstu.edu

†These authors contributed equally to this work

Competing interest: The authors declare that no competing interests exist.

## Abstract

Previously, it was shown that learned avoidance of *Pseudomonas aeruginosa* PA14 in *Caenorhabditis elegans* can be transmitted to untrained offspring (Kaletsky et al., 2025). Here, we show an independent validation of this phenomenon under the originally specified assay conditions. Adapting the Murphy lab protocol, worms trained on PA14 develop significant avoidance that persists in F1 and F2 generations, with reduced magnitude after P0. These results align with the Murphy group's findings and address a recent report from the Hunter group that did not detect persistence beyond F1.

## Introduction

*Caenorhabditis elegans* inhabits soil environments where it encounters diverse bacteria, including both nutritional and pathogenic species (*Schulenburg and Félix, 2017*; *Neher, 2010*). This nematode has evolved sophisticated sensory mechanisms to distinguish beneficial from harmful bacteria (*Diaz et al., 2015*; *Kim and Flavell, 2020*; *Guillermin, 2018*). Interestingly, naïve *C. elegans* initially prefer the pathogenic bacterium *Pseudomonas aeruginosa* (PA14) over the nonpathogenic laboratory strain *Escherichia coli* (OP50) but rapidly learn to avoid PA14 following exposure (*Zhang et al., 2005*).

The Murphy lab first demonstrated that this learned avoidance behavior can be transmitted to progeny, persisting through the F2 generation (*Baugh and Day, 2020*; *Moore et al., 2019*; *Kaletsky et al., 2020*; *Moore, 2021*; *Sengupta et al., 2024*). This transgenerational epigenetic inheritance has been linked to small RNA pathways and the dsRNA transport proteins SID-1 and SID-2 (*Cecere, 2021*). Recently, *Sengupta et al., 2024*, further identified a specific small RNA responsible for inducing transgenerational inheritance of learned avoidance.

**Figure 1.** Transgenerational inheritance of PA14 avoidance in *C. elegans*. (**A**) Chemotaxis assay setup. A 25 µL drop of OP50 and PA14 bacteria of the same optical density (OD$_{600}$=1) plus 1.0 µL of 400 mM NaN$_3$ were placed equidistant from the center of a 10 cm chemotaxis plate where worms were placed at the start of the assay (see Materials and methods). Indices were measured for naïve worms, trained worms, first-generation (**F1**), and second-generation (**F2**) offspring. (**B**) Pooled results from four independent trials showing that trained worms, which were exposed to PA14 for 24 hr, exhibited significantly higher choice indices, indicating learned avoidance of PA14. This aversion persisted in F1 and F2 progeny, demonstrating transgenerational inheritance of pathogen avoidance behavior. (**C**) Comparison of data separated into individual trials. Each point is one plate assay. Bars show mean ± s.e.m. Sample sizes (assays): Naïve n = 38, Trained n = 38, F1 n = 19, F2 n = 43. Datasets that met normality and equal variance were analyzed by one-way ANOVA with Holm–Sidak post hoc comparisons versus Naïve; otherwise Kruskal–Wallis with Dunn's post hoc was used. Significance versus Naïve is indicated as *p < 0.05 or **p < 0.01, as reported in Supplementary file (see ***Supplementary file 1A*** for statistics and ***Supplementary file 1B*** for raw data).

However, ***Gainey et al., 2025***, representing the Hunter group, reported that while parental and F1 avoidance behaviors were evident, transgenerational inheritance was not reliably observed beyond the F1 generation under their experimental conditions. Critically, ***Kaletsky et al., 2025***, demonstrated that omission of sodium azide during scoring can completely abolish detection of inherited avoidance, revealing that this key methodological difference may explain the conflicting results between laboratories. The use of sodium azide to immobilize worms at the moment of initial bacterial choice appears essential for capturing the inherited behavioral response. These findings highlight how seemingly minor methodological variations can dramatically impact detection of transgenerational inheritance and underscore the need for independent replication using standardized protocols. Here, we adapted the protocol established by the Murphy group, maintaining the critical use of sodium azide to paralyze worms at the time of choice, to test whether parental exposure to PA14 elicits consistent avoidance in subsequent generations. Our study specifically focuses on the transmission of learned avoidance through the F2 generation, beyond the intergenerational (F1) effect, because this is where divergence between published studies begins. Our goal was to evaluate whether the reported transgenerational signal persists across two generations without reinforcement.

## Results

We investigated whether *C. elegans* exposed to PA14 exhibit learned avoidance behavior that persists across generations. Naïve worms showed a preference for PA14 over OP50, with a negative choice index (approximately –0.4). After 24 hr of exposure to PA14, trained worms developed strong aversion to the pathogen, exhibiting a significantly positive choice index (***Figure 1***).

Critically, this avoidance behavior persisted in the untrained F1 and F2 progeny of trained worms, with both generations showing significantly higher choice indices compared to naïve controls (***Figure 1*** and ***Supplementary file 1***).

## Discussion

Our findings independently validate the transgenerational inheritance of learned pathogen avoidance initially reported by the Murphy lab. In contrast, ***Gainey et al., 2025***, using a modified protocol, failed to observe such inheritance beyond F1 despite multiple attempts. While we cannot definitively reconcile these differences, our results suggest that under tightly controlled conditions, including bacterial lawn density, OD$_{600}$=1.0 standardization, and immobilization with sodium azide restricted to 1 hr (to

capture initial preference behaviors), transgenerational inheritance through F2 is both detectable and statistically robust.

This underscores the assay's sensitivity to environmental variables, such as synchronization method and bacterial lawn density. This highlights the importance of consistency across experimental setups and supports the view that context-dependent variation may underlie previously reported discrepancies.

These results confirm that, under tightly controlled conditions, transgenerational inheritance of PA14 avoidance extends through the F2 generation, though the response attenuates without further reinforcement.

## Materials and methods

### Nematode strains and maintenance

Wild-type *C. elegans* (N2) were maintained on nematode growth medium (NGM) plates seeded with *E. coli* OP50 at 20°C under standard conditions (*Brenner, 1974*). Worm populations were passaged regularly to avoid overcrowding and starvation. OP50 was obtained from the Caenorhabditis Genetics Center (CGC). Please refer to *Supplementary file 1C* for reagents used in this study, which are presented side by side with the lists of reagents supplied by the Hunter and Murphy groups.

### Bacterial cultures

We inoculated 6 mL of LB medium (pH 7.5) with single colonies of OP50 or *P. aeruginosa* PA14 and incubated them overnight at 37°C with shaking at 250 rpm. For PA14, cultures did not exceed 16 hr of incubation or develop visible biofilms. For both OP50 and PA14, the overnight culture was diluted to an $OD_{600}$ of 1.0 in fresh LB media prior to seeding.

### Synchronization of worms

Gravid adults were collected from OP50-seeded NGM plates by washing with liquid NGM buffer. We used liquid NGM buffer instead of M9 buffer (as specified in the original Murphy protocol) to maintain chemical consistency with the solid NGM culture plates. This modification minimizes potential osmotic stress since liquid NGM matches the pH (6.0) and ionic composition of the growth medium, whereas M9 buffer has a different pH (7.0) and ionic profile. After settling by gravity, a standard bleach solution was added and gently nutated for 5–10 min. The embryo pellet was washed two to three times with liquid NGM after centrifugation. Liquid NGM buffer is identical to agar NGM plates except that it does not include agar, peptone, cholesterol, or OP50 *E. coli*. The key difference between liquid NGM and M9 (aside from their pH) is that NGM has $CaCl_2$ and that M9 typically has more $Na^+$ while NGM has more $K^+$ due to the phosphate salts used. Approximately 250–350 eggs were plated onto each OP50-seeded NGM plate and incubated at 20°C for 48–52 hr until reaching late L4 stage.

### Training and choice assay

Training plates (10 cm NGM) were seeded with 1 mL of overnight bacterial cultures and incubated at 25°C for 2 days. Choice assay plates (6 cm NGM) were prepared with 25 µL spots of OP50 and PA14 on opposite sides placed 24 hr prior to the assays. Late L4 worms were transferred to training plates (OP50 or PA14) for 24 hr at 20°C. For choice assays, 1.0 µL of 400 mM sodium azide was placed on each bacterial spot before adding worms. After 1 hr, the number of worms immobilized on each spot was counted to calculate a choice index (CI):

$$CI = (\text{worms on OP50} - \text{worms on PA14}) / (\text{worms on OP50} + \text{worms on PA14})$$

We used 400 mM sodium azide rather than the 1 M concentration reported by *Moore et al., 2019*, because preliminary trials showed that higher concentrations caused premature paralysis before worms could reach either bacterial spot, potentially biasing choice measurements. The 400 mM concentration provided sufficient immobilization while preserving the behavioral choice window.

### Transgenerational testing

Synchronized F1 progeny were obtained by bleaching trained adult worms and allowing embryos to develop on OP50-seeded plates at 20°C. On day 1 of adulthood, F1 worms were tested using the

same choice assay. The F2 generation was derived from untrained F1 adults, synchronized in the same manner, and tested in identical conditions.

Each behavioral assay was conducted using animals from a biologically independent growth plate. While F2 plates were derived from pooled embryos from multiple F1 parents, each assay represents an independent biological replicate with no reuse of animals across assays. F2 assays (n=45) exceeded F1 assays (n=20) due to PA14-induced fecundity reduction in trained worms, limiting the number of viable F1 progeny. The higher number of F2 assays reflects the greater reproductive success of healthy F1 animals and provides additional statistical power for population-level behavioral comparisons.

## Controls and additional considerations

The data presented in this study are the result of four independent rounds of experimentations. Each individual assay was performed with animals harvested from unique culture plates, ensuring biological independence across all behavioral measurements. An average of 62±43 animals participated in each assay (were immobilized in the $NaN_3$ and tallied at the end of the assay). We conducted an average of 8.5 assays per condition during each of the four replicates. Each replicate was a biologically independent experiment conducted on a different day. Our experimental design employed population-level comparisons across generations using unpaired statistical analyses, with no attempt to track individual lineages across generations. All plates used in the study were prepared at least 2 days before experiments to ensure consistent bacterial lawn growth. Worms were monitored to prevent starvation or overcrowding, as these conditions could influence their bacterial preferences and behavioral responses. Fresh bacterial stocks were maintained at –80°C and streaked weekly to ensure culture consistency.

## Statistical analysis

All statistical analyses were performed using SigmaPlot 14 (Inpixon). Population-level behavioral scoring follows established practice for high-throughput *C. elegans* assays (*Swierczek et al., 2011*). Outliers were identified and removed using the interquartile range criterion (>1.5 × IQR), resulting in the exclusion of 5 out of 143 data points (3.5%). Their removal did not alter the significance or direction of reported effects. When datasets passed both the Shapiro-Wilk test for normality and the Brown-Forsythe test for equal variance, we used one-way analysis of variance (ANOVA) followed by Holm-Sidak post hoc comparisons versus the control group. For datasets that failed either assumption, we used Kruskal-Wallis one-way ANOVA on ranks, followed by Dunn's post hoc comparisons versus control. The full statistical outputs and raw data for individual assays are presented in *Supplementary file 1*.

## Acknowledgements

We thank Dr. Murphy at Princeton University for reagents and protocols. Some strains were provided by the CGC, which is funded by NIH Office of Research Infrastructure Programs (P40 OD010440). Funding was provided by the National Institutes of Health, National Institute of Arthritis and Musculo-skeletal and Skin Diseases (award 2R15AR068583-02).

## Additional information

### Funding

| Funder | Grant reference number | Author |
| --- | --- | --- |
| National Institute of Arthritis and Musculoskeletal and Skin Diseases | 2R15AR068583-02 | Andres Gabriel Vidal-Gadea |

The funders had no role in study design, data collection and interpretation, or the decision to submit the work for publication.

## Author contributions
Aalimah Akinosho, Data curation, Formal analysis, Investigation, Methodology, Writing – original draft; Joseph Alexander, Resources, Methodology; Kyle Floyd, Resources, Supervision, Methodology; Andres Gabriel Vidal-Gadea, Conceptualization, Resources, Data curation, Formal analysis, Supervision, Funding acquisition, Methodology, Writing – original draft, Project administration, Writing – review and editing

## Author ORCIDs
Andres Gabriel Vidal-Gadea https://orcid.org/0000-0001-5981-5528

Reviewer #1 (Public review): https://doi.org/10.7554/eLife.107034.3.sa1
Reviewer #2 (Public review): https://doi.org/10.7554/eLife.107034.3.sa2
Reviewer #3 (Public review): https://doi.org/10.7554/eLife.107034.3.sa3
Author response https://doi.org/10.7554/eLife.107034.3.sa4

## Additional files

### Supplementary files
Supplementary file 1. Full statistics, raw chemotaxis assay counts, and reagent lists for all experiments. (**A**) Statistical analyses of chemotaxis choice indices across generations. Results from Kruskal-Wallis one-way analysis of variance (ANOVA) followed by Dunn's or Holm-Sidak post hoc comparisons are reported for naïve, trained, F1, and F2 animals. For each trial, test statistics (H, DF, p values) and pairwise comparisons are presented. Normality (Shapiro-Wilk) and equal variance (Brown-Forsythe) test outcomes are included. (**B**) Raw chemotaxis assay data for all replicates. Counts of worms immobilized on PA14 and OP50 spots are listed for each assay, along with calculated choice index (CI) values. Data are separated by generation (naïve, trained, F1, F2) and trial number, providing the full dataset underlying *Figure 1*. (**C**) Reagents used in this study compared with those supplied by the Murphy and Hunter groups. The table lists reagents, suppliers, and catalog numbers for nematode growth medium (NGM) ingredients and related chemicals. Parallel listing highlights methodological consistencies and differences across the three laboratories.

MDAR checklist

### Data availability
*Supplementary file 1* contains the results and summary of data acquired in this study.

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
