## [Editor Report · eLife Assessment]

This **valuable** study concerns a model for transgenerational epigenetic inheritance, the learned avoidance by *C. elegans* of the PA14 pathogenic strain of *Pseudomonas aeruginosa*. A recent study questioned whether transgenerational inheritance in this paradigm lacks robustness. The authors of this study have worked independently of the group that reported the original phenomenon and also independently of the group that challenged the original report. With **solid** data, this study independently validates findings previously reported by the Murphy group, confirming that the paradigm is reproducible elsewhere. The reviewers also appreciated the information on reagent sources used by different groups. The present study is therefore of broad interest to anyone studying genetics, epigenetics, or learned behavior.

---

## [Referee Report · Reviewer #1 (Public review)]

Summary:

The manuscript addresses the discordant reports of the Murphy (Moore et al., 2019; Kaletsky et al., 2020; Sengupta et al., 2024) and Hunter (Gainey et al., 2025) groups on the existence (or robustness) of transgenerational epigenetic inheritance (TEI) controlling learned avoidance of *C. elegans* to *Pseudomonas aeruginosa*. Several papers from Colleen Murphy's group describe and characterize *C. elegans* transgenerational inheritance of avoidance behaviour. In the hands of the Murphy group, the learned avoidance is maintained for up to four generations, however, Gainey et al. (2025) reported an inability to observe inheritance of learned avoidance beyond the F1 generation. Of note, Gainey et al used a modified assay to measure avoidance, rather than the standard assay used by the Murphy lab. A response from the Murphy group suggested that procedural differences explained the inability of Gainey et al.(2025) to observe TEI. They found two sources of variability that could explain the discrepancy between studies: the modified avoidance assay and bacterial growth conditions (Kaletsky et al., 2025). The standard avoidance assay uses azide as a paralytic to capture worms in their initial decision, while the assay used by the Hunter group does not capture the worm's initial decision but rather uses cold to capture the location of the population at one point in time.

In this short report, Akinosho, Alexander, and colleagues provide independent validation of transgenerational epigenetic inheritance (TEI) of learned avoidance to *P. aeruginosa* as described by the Murphy group by demonstrating learned avoidance in the F2 generation. These experiments used the protocol described by the Murphy group, demonstrating reproducibility and robustness.

Strengths:

Despite the extensive analyses carried out by the Murphy lab, doubt may remain for those who have not read the publications or for those who are unfamiliar with the data, which is why this report from the Vidal-Gadea group is so important. The observation that learned avoidance was maintained in the F2 generation provides independent confirmation of transgenerational inheritance that is consistent with reports from the Murphy group. It is of note that Akinosho, Alexander et al. used the standard avoidance assay that incorporates azide, and followed the protocol described by the Murphy lab, demonstrating that the data from the Moore and Kaletsky publications are reproducible, in contrast to what has been asserted by the Hunter group.

Comments on revised version:

I am happy with the responses to reviews.

---

## [Referee Report · Reviewer #2 (Public review)]

Summary:

The manuscript "Independent validation of transgenerational inheritance of learned pathogen avoidance in *C. elegans*" by Akinosho and Vidal-Gadea offers evidence that learned avoidance of the pathogen PA14 can be inherited for at least two generations. In spite of initial preference for the pathogen when exposed in a 'training session', 24 hours of feeding on this pathogen evoked avoidance. The data are robust, replicated in 4 trials, and the authors note that diminished avoidance is inherited in generations F1 and F2.

Strengths:

These results contrast with those reported by Gainey et al, who only observed intergenerational inheritance for a single generation. Although the authors' study does not explain why Gainey et el fail to reproduce the Murphy lab results, one possibility is that a difference in a media ingredient could be responsible.

Comments on revised version:

The responses to the reviewer comments appear reasonable for the most part.

---

## [Referee Report · Reviewer #3 (Public review)]

Summary:

This short paper aims to provide an independent validation of the transgenerational inheritance of learned behaviour (avoidance) that has been published by the Murphy lab. The robustness of the phenotype has been questioned by the Hunter lab. In this paper, the authors present one figure showing that transgenerational inheritance can be replicated in their hands. Overall, it helps to shed some light on a controversial topic.

Strengths:

The authors clearly outline their methods, particularly regarding the choice of assay, so that attempting to reproduce the results should be straightforward. It is nice to see these results repeated in an independent laboratory.

Comments on revised version:

I'm happy with the response to reviewers.

---

## [Author Response]

The following is the authors’ response to the original reviews.

**Reviewer #1 (Public Review):**
Confirmation of daf-7::GFP data and inheritance beyond F2

Reviewer suggested confirming daf-7::GFP molecular marker data and testing inheritance beyond the F2 generation to further strengthen the findings.

We agree these experiments would provide valuable mechanistic insights into the molecular basis of transgenerational inheritance. However, our study was specifically designed as a reproducibility study focusing on the central controversy regarding F2 inheritance (Gainey et al. vs. Murphy lab findings). The daf-7::GFP molecular marker experiments, while important for understanding mechanisms, represent a different research question requiring extensive additional resources and expertise beyond the scope of this validation study. Our primary goal was to provide independent confirmation of the disputed F2 inheritance using standardized behavioral assays. It is our hope that future work will pursue these important mechanistic validations.

"Exhaustive attempts" language

Reviewer disagreed with characterizing Gainey et al.'s efforts as "exhaustive attempts" since they modified the original protocol.

We revised this statement in the Results and Discussion to more accurately reflect the experimental situation: "In contrast, Gainey et al. (2025), representing the Hunter group, reported that while parental and F1 avoidance behaviors were evident, transgenerational inheritance was not reliably observed beyond the F1 generation under their experimental conditions."

Importance of sodium azide

Reviewer suggested including more discussion about the recent findings on the importance of sodium azide in the assay, referencing the Murphy group's response paper.

We have prominently highlighted the critical role of sodium azide in our Introduction with strengthened language that emphasizes its importance for resolving the scientific controversy: "Critically, Kaletsky et al. (2025) demonstrated that omission of sodium azide during scoring can completely abolish detection of inherited avoidance, revealing that this key methodological difference may explain the conflicting results between laboratories. The use of sodium azide to immobilize worms at the moment of initial bacterial choice appears essential for capturing the inherited behavioral response. These findings highlight how seemingly minor methodological variations can dramatically impact detection of transgenerational inheritance and underscore the need for independent replication using standardized protocols."

Protocol fidelity statement

Reviewer requested a more direct statement clarifying that we followed the Murphy group protocol, noting that we made some modifications.

We followed the core Murphy lab protocol with two evidence-based optimizations that preserve the essential experimental elements: (1) We used 400 mM sodium azide instead of 1 M based on preliminary data showing the higher concentration caused premature paralysis before worms could make behavioral choices, and (2) We used liquid NGM buffer instead of M9 to maintain chemical consistency with the solid NGM plates used for worm culture, minimizing potential osmotic stress. These modifications improved experimental reliability while maintaining the critical components: sodium azide immobilization, bacterial lawn density standardization (OD_600_ = 1.0), and synchronized scoring conditions that are essential for detecting inherited avoidance.

Overstated dilution claim

Reviewer noted that the statement about "gradual decrease" in avoidance strength was overstated and didn't reflect the actual data presented in the manuscript.

We removed this statement.

Environmental variables phrasing

Reviewer found the sentence about environmental variables unclear, noting that Gainey et al. didn't actually acknowledge variability but saw it as indicating error or stochastic processes.

We refined this statement for greater precision and clarity: "This underscores the assay's sensitivity to environmental variables, such as synchronization method and bacterial lawn density. This highlights the importance of consistency across experimental setups and support the view that context-dependent variation may underlie previously reported discrepancies."

**Reviewer #2 (Public Review):**
Reagent sourcing

Reviewer suggested listing the sources of media ingredients with company names and catalog numbers, as this might be important for reproducibility.

To ensure complete reproducibility, we created a comprehensive Table S3 listing all reagents, suppliers, and catalog numbers used in our experiments. This detailed information enables exact replication of our experimental conditions and addresses potential variability that might arise from different reagent sources between laboratories.

**Reviewer #3 (Public Review):**
Raw data transparency

Reviewer noted that while a spreadsheet with choice assay results was provided, the individual raw data from assays was not included, which would be helpful for assessing sample sizes.

We now provide complete experimental transparency through Table S2, which contains individual choice indices from all 138 assays conducted across four independent trials. This comprehensive dataset allows full assessment of our experimental outcomes, statistical robustness, and reproducibility while enabling other researchers to perform independent statistical analyses.

F1/F2 assay disparity

Reviewer questioned whether the higher number of F2 assays compared to F1 represented truly independent assays, asking if multiple F2 assays were performed from offspring of one F1 plate (which would not represent independent assays).

We clarified this important statistical consideration in Methods (Transgenerational Testing): "Each behavioral assay was conducted using animals from a biologically independent growth plate. While F2 plates were derived from pooled embryos from multiple F1 parents, each assay represents an independent biological replicate with no reuse of animals across assays. F2 assays (n=45) exceeded F1 assays (n=20) due to PA14-induced fecundity reduction in trained worms, limiting the number of viable F1 progeny. The higher number of F2 assays reflects the greater reproductive success of healthy F1 animals and provides additional statistical power for population-level behavioral comparisons." We also enhanced our Controls section to clarify that "Our experimental design employed population-level comparisons across generations using unpaired statistical analyses, with no attempt to track individual lineages across generations."

Methodological variations overstatement

Reviewer felt the Introduction overstated the findings by suggesting the authors "address potential methodological variations," when they only used one assay setup throughout.

We have corrected the Introduction to accurately reflect our study design and scope: "Here, we adapted the protocol established by the Murphy group, maintaining the critical use of sodium azide to paralyze worms at the time of choice, to test whether parental exposure to PA14 elicits consistent avoidance in subsequent generations. Our study specifically focuses on the transmission of learned avoidance through the F2 generation, beyond the intergenerational (F1) effect, because this is where divergence between published studies begins."

**Reviewer #1 (Recommendations for the authors):**
Worm numbers

Reviewer noted that information about the number of worms used should be included in the training and choice assay methods section rather than separated.

We clarified worm numbers and sample sizes in the Methods (Controls and Additional Considerations): "Each individual assay averaged 62 ± 43 animals (range: 15-150 worms per assay), with a total of 138 assays conducted across four independent experimental trials. The variation in worm numbers per assay reflects natural variation in worm recovery and immobilization efficiency during choice assays. We conducted an average of 8.5 assays per condition during each of the four replicates."

Figure 1 legend and consistency

Reviewer identified several issues: inconsistent terminology ("treated" vs "trained"), incorrect statistical test naming, missing p-value annotations, and need for consistency between figure and legend. We have systematically addressed all figure consistency and statistical annotation issues:

Replaced inconsistent "treated" terminology with "trained" throughout

Corrected the statistical test description to accurately reflect our analysis: "Kruskal-Wallis oneway ANOVA followed by Dunn's post hoc" which properly corresponds to the statistical tests detailed in Table S1

Added explicit p-value annotations in the figure legend: "*p<0.05, **p<0.01 means and SEM shown (see Table S1 for statistics and Table S2 for raw data)"

Ensured consistent terminology between figure and legend

NGM vs. M9 buffer

Reviewer questioned whether we used NGM buffer or M9 buffer for washing steps, noting that NGM isn't usually referred to as "buffer."

We have prominently featured and thoroughly clarified our rationale for using liquid NGM buffer in the Methods (Synchronization of Worms section). The explanation now appears upfront in the methods: "We used liquid NGM buffer instead of M9 buffer (as specified in the original Murphy protocol) to maintain chemical consistency with the solid NGM culture plates. This modification minimizes potential osmotic stress since liquid NGM matches the pH (6.0) and ionic composition of the growth medium, whereas M9 buffer has a different pH (7.0) and ionic profile." We provide detailed chemical differences and explain that this modification maintains consistency with culture conditions while preserving essential experimental procedures.

Grammar/typos

Reviewer noted that the manuscript needed thorough proofreading to address grammatical errors and typographical mistakes.

We have conducted comprehensive proofreading and editing throughout the manuscript to resolve grammatical and typographical errors. Specific improvements include: clarified sentence structure in the Introduction and Results sections, corrected technical terminology consistency, improved figure legend clarity, and enhanced overall readability while maintaining scientific precision.

Sodium azide concentration

Reviewer noted that our sodium azide concentration differed from the Moore paper and requested comment on this difference.

We have included explicit justification for our sodium azide concentration choice in the Methods (Training and Choice Assay): "We used 400 mM sodium azide rather than the 1 M concentration reported by Moore et al. (2019) because preliminary trials showed that higher concentrations caused premature paralysis before worms could reach either bacterial spot, potentially biasing choice measurements. The 400 mM concentration provided sufficient immobilization while preserving the behavioral choice window."

**Reviewer #2 (Recommendations for the authors):**
Comparative reagent analysis

Reviewer suggested creating a supplemental table comparing reagent sources between our study, Gainey et al., and Murphy et al., proposing that media ingredient differences might explain the discrepancies.

While direct reagent comparison between laboratories was beyond the scope of this validation study, we recognize this as an important consideration for understanding experimental variability. Our comprehensive reagent sourcing information (Table S3) provides the foundation for future comparative studies. We encourage collaborative efforts to systematically compare reagent sources across laboratories, as media component differences could contribute to the experimental variability observed between research groups. Such analyses would be valuable for establishing standardized protocols across the field.

Conclusion

We hope that these revisions satisfactorily address the reviewers’ concerns. We believe these improvements significantly strengthened the manuscript's contribution to resolving this important scientific controversy.

We thank the reviewers again for their invaluable insights and constructive feedback, which have substantially improved the quality and impact of our work.